

# *Bracon* wasps for ecological pest control–a laboratory experiment

Jessica Lettmann[1], Karsten Mody[1,2], Tore-Aliocha Kursch-Metz[3],
Nico Blüthgen[1] and Katja Wehner[1]

[1] Ecological Networks, Technische Universität Darmstadt, Darmstadt, Germany
[2] Department of Applied Ecology, Hochschule Geisenheim University, Geisenheim, Germany
[3] AMW Nützlinge GmbH, Darmstadt, Germany

Corresponding author
Katja Wehner, kdwehner@gmx.de

## ABSTRACT

Biological control of pest insects by natural enemies may be an effective, cheap and environmentally friendly alternative to synthetic pesticides. The cosmopolitan parasitoid wasp species *Bracon brevicornis* Wesmael and *B. hebetor* Say (Hymenoptera: Braconidae) use lepidopteran species as hosts, including insect pests like *Ephestia kuehniella* or *Ostrinia nubilalis*. Here, we compare the reproductive success of both *Bracon* species on *E. kuehniella* in a laboratory experiment. We asked (1) how the reproductive success on a single host larva changes with temperature, (2) how it changes with temperature when more host larvae are present and (3) how temperature and availability of host larvae influence the efficacy of *Bracon* species as biological control agents. In general, differences between *B. brevicornis* and *B. hebetor* have been small. For rearing both *Bracon* species in the laboratory on one host larva, a temperature between 20–27 °C seems appropriate to obtain the highest number of offspring with a female-biased sex ratio. Rearing the braconid wasps on more than one host larva revealed a higher number of total offspring but less offspring per host larva on average. Again, highest numbers of offspring hatched at 27 °C and the sex ratio was independent from temperature. Although no parasitoids hatched at 12 °C and only few at 36 °C, host larvae were still paralyzed. The efficacy of *B. brevicornis* was higher than 80% at all numbers of host larvae presented at all temperatures while the efficacy of *B. hebetor* was less than 80% at 12 °C and 27 °C at low numbers of host larvae presented. In conclusion, practitioners can use either *B. brevicornis* or *B. hebetor* at low and high temperatures and at varying host densities to achieve high pest control efficacy.

## INTRODUCTION

Intensified land-use often causes a loss of biodiversity of above- and below-ground organisms and a homogenization of plant and animal communities by the reduction of habitat diversity or habitat destruction (*Andow, 1983*; *Hendrickx et al., 2007*; *Weiner et al., 2011*; *Allan et al., 2015*; *Newbold et al., 2015*; *Chisté et al., 2016*; *Gossner et al., 2016*). Such losses of biodiversity affect ecosystem services; e.g., the loss of pollinator species such as wild bees or bumblebees causes a reduction in pollination (*Biesmeijer et al., 2006*; *Potts*

*et al., 2010*). Loss of habitats and an alteration of landscapes also impair the service of natural pest control by their natural enemies (*Kruess & Tscharntke, 1994*; *Bianchi, Booij & Tscharntke, 2006*; *Rusch et al., 2013*; *Pinero & Manandhar, 2015*) resulting in the spread of insect pests (*Balzan, Bocci & Moonen, 2016*; *Hajek & Eilenberg, 2018*). A prominent example of a pest insect that is spread over many European countries and also large parts of North America is the European corn borer *Ostrinia nubilalis* (Hübner) (Lepidoptera: Crambidae), which profits from corn monocultures and causes high yield losses (*Štěpánek, Veselá & Muška, 2014*). The moths' larvae burrow into the stem, disrupt the nutrient supply and destabilize the plant (*Capinera, 2000*; *Vétek et al., 2017*). Natural enemies like the insidious flower bug *Orius insidiosus* (Say) (Hemiptera: Anthocoridae) or parasitoid wasps are less abundant in monocultures than in diverse landscapes (*Lundgren, Wyckhuys & Desneux, 2009*; *Pak et al., 2015*).

Synthetic insecticides are commonly used to prevent damages caused by insect pests in granaries and in agricultural landscapes. Such pesticides are often harmful and toxic to aquatic organisms, pollinators (*Hallberg, 1989*; *Dunier & Siwicki, 1993*; *Arias-Estévez et al., 2008*) as well as to natural antagonists of pests (*Ruberson, Nemoto & Hirose, 1998*). Alternatively, biological pest control uses biological agents for the control of different pests. These biological control agents need to be easily applicable, and should have low production costs while being effective at the same time. Biological pest control can be achieved by antagonists of insect pests (*Kremen & Chaplin-Kramer, 2007*) or pathogens like bacteria or viruses (*Lacey et al., 2015*). The use of parasitoid wasps has been shown to be an efficient method for controlling insect pests in agricultural systems and stored products (e.g., *Schöller et al., 2018*; *Wang et al., 2019*).

Many organisms already used as biological control agents belong to the family Braconidae (Hymenoptera), including the species *Bracon (=Habrobracon) brevicornis* Wesmael and *Bracon (=Habrobracon) hebetor* Say. Due to their promising use in biological pest control, both *Bracon* species have been investigated for decades (e.g., *King, 1987*; *Galloway & Grant, 1988*; *Zaki et al., 1998*; *Akman Gündüz & Gülel, 2005*; *Saadat et al., 2016*). The *Bracon* species are cosmopolitan, polyphagous, gregarious, larval ectoparasitoids (*Antolin, Ode & Strand, 1995*). They use different lepidopteran species as hosts including many insect pests like *Ephestia kuehniella* Zeller (Lepidoptera: Pyralidae), *Galleria mellonella* (Linnaeus) (Lepidoptera: Pyralidae), *Corcyra cephalonica* (Stainton) (Lepidoptera: Pyralidae), *Plodia interpunctella* (Hübner) (Lepidoptera: Pyralidae) or *Ectomyelois ceratoniae* (Zeller) (Lepidoptera: Pyralidae) (*Ghimire & Phillips, 2010*; *Saadat, Bandani & Dastranj, 2014*; *Farag et al., 2015*; *Khalil et al., 2016*; *Singh, Singh & Tripathi, 2016*; *Amadou et al., 2019*). By stinging the host larva the parasitoid wasp injects a venom and paralyzes the host larva, which stops feeding and moving immediately (*Abbas, 1980*; *Wührer & Zimmermann, 2008*; *Wyss, Wührer & Zimmermann, 2010*). Afterwards, eggs are laid on the outside of the host larva, which remains paralyzed. After hatching, the wasp larvae feed on their host and finally kill it (Fig. 1).

The Mediterranean flour moth *Ephestia kuehniella* is a cosmopolitan pest of stored grain products (*Jacob & Cox, 1977*; *Ayvaz & Karabörklü, 2008*; *Xu, Wang & He, 2008*). By spinning webbings that clog the machinery in industrial flourmills, *E. kuehniella* causes

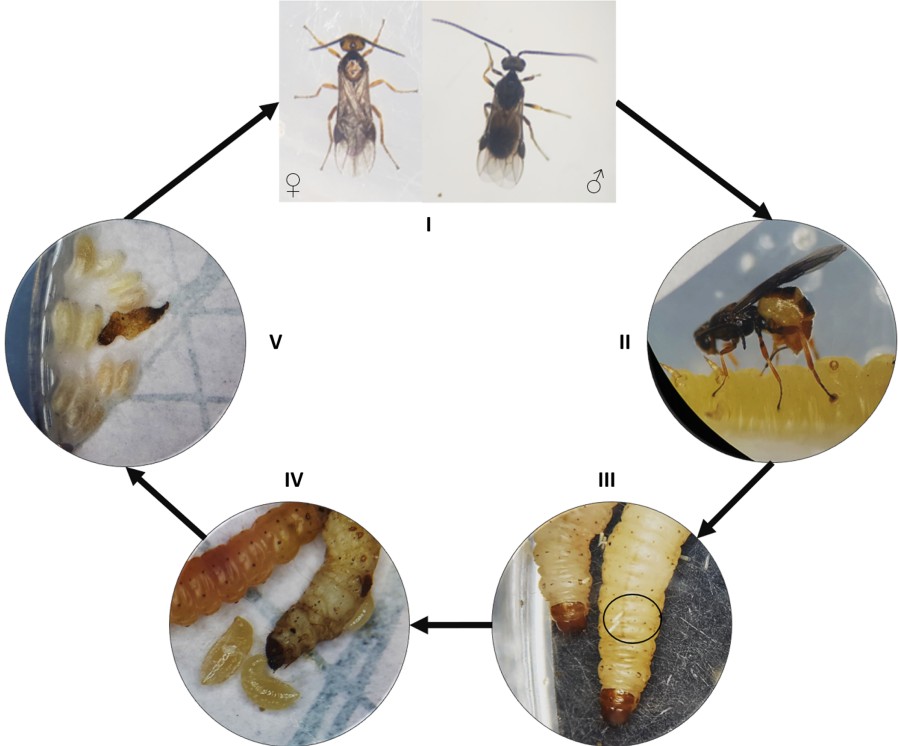

**Figure 1 Life cycle of *Bracon brevicornis* and *B. hebetor*.** Life cycle of *Bracon brevicornis* and *B. hebetor*. I: female and male wasps, II: female wasp injects venom, III: parasitoid eggs on host (encircled), IV: parasitoid larvae, V: pupated parasitoid larvae.           

economic damages (*Jacob & Cox, 1977*; *Ayvaz & Karabörklü, 2008*). Eggs or larvae of *E. kuehniella* are commonly used to rear parasitoids such as *Trichogramma* or *Bracon* (*Faal-Mohammad-Ali & Shishehbor, 2013*; *Xu, Wang & He, 2008*; *St-Onge et al., 2015*).

    *Bracon brevicornis* and *B. hebetor* are promising species for biological pest control since females lay eggs on various pest host species, stop their development by paralyzing them and finally reduce pest densities (*Taylor, 1988*; *Akman Gündüz & Gülel, 2005*). However, both *Bracon* species are very similar morphologically (separated by wing venation, larval morphology and genital characters; *Matthews, 1974*) and also show similar life cycles (*Alam et al., 2016*; *Srinivasan & Mohan, 2017*). This parasitic life cycle can be influenced by several factors (e.g., *Saadat et al., 2014*). There is evidence that *Bracon* females can regulate clutch size based on the size and/or quality of the host (*Taylor, 1988*; *Godfray, 1994*; *Milonas, 2005*). Furthermore, host density and density of the parasitoid population seem to influence life cycle statistics and sex ratios; i.e., *B. hebetor* produced more females if densities were high (*Galloway & Grant, 1988*; *Singh, Singh & Tripathi, 2016*). Additionally, temperature plays an important role in the developmental biology of *Bracon* species and also affects the efficacy of parasitization (*Rao & Kumar, 1960*; *Thanavendan & Jeyarani, 2010*). As agents for biological control, braconid wasps are often used in addition to trichogrammatid parasitoids which parasitize eggs of insect pests (*Brower, 1988*) which is also done to dam *E. kuehniella* (*Ayvaz & Karabörklü, 2008*; *St-*

*Onge et al., 2015*). If high pest pressure occurs and many eggs remain unparasitized and develop to larvae, *B. brevicornis* and *B. hebetor* can be used in support.

Ideally, the rearing of biological control agents should be inexpensive and simple, with no special requirements or culture conditions. Furthermore, in *Bracon* wasps a high amount of female offspring is desired since only female wasps paralyze and kill the host larvae. For practitioners, high efficacy, i.e., a high number of paralyzed and killed pest larvae, at low cost is important. To further investigate and compare the best rearing conditions and efficacy of *B. brevicornis* and *B. hebetor*, we asked:

(1) how reproductive success of female wasps on a single host *Ephestia kuehniella* larva changes with temperature,

(2) how reproduction success of female wasps changes with the availability of more host larvae at different temperatures, and

(3) how temperature and varying availability of hosts affect the number and mortality of paralyzed host larvae and therefore the potential efficacy of braconid wasps as biocontrol agents.

## MATERIALS AND METHODS

### Rearing of *Bracon brevicornis* and *Bracon hebetor*

The laboratory studies were conducted between January and June 2019. The first generation of *B. brevicornis* and *B. hebetor* was provided by AMW Nützlinge, Pfungstadt, Germany. Cultivation started with three strains of *B. hebetor* and two strains of *B. brevicornis* collected from different hosts (Appendix 1). For rearing, two females of either *B. hebetor* or *B. brevicornis* and ten *Ephestia kuehniella* larvae (last stage before pupation L6) were placed in Petri dishes with 5 cm diameter, using five to ten Petri dishes per strain and generation. Until pupation, Petri dishes were placed in climate chambers at 27 °C, 70% relative humidity (RH) and a light-dark photoperiod of 16:8 (L:D). Subsequently, the cocoons were transferred in glass bottles and left at room temperature until hatching. Female and male wasps were kept together after hatching and fed with sugar water. At the earliest 48 h after hatching but not later than 5 days after hatching, the female wasps were used for the experiments. Unused specimens were frozen after 5 days, and only young females were ever used.

### Reproductive success of female wasps on a single host larva at different temperatures

The number of offspring from each individual female wasp of either *B. brevicornis* or *B. hebetor* on one larva of *E. kuehniella* was recorded. Therefore, one female of each *Bracon* strain and one host larva were set in a Petri dish with 5 cm diameter and placed in climate chambers with 12 °C, 20 °C, 27 °C or 36 °C with steady humidity of 70% and 16:8 L:D photoperiod. Each temperature was replicated per species and strain; the final number of replications varied due to difference in the developmental cycle and the survival rate (see Appendix 2 for initial numbers of replications). Samples were frozen when all wasps

had hatched, but no later than 5 weeks after the start of the experiment. Subsequently, the number of hatched female and male wasps was determined.

## Reproductive success of female wasps on variable numbers of host larvae at different temperatures

The number of offspring of one female wasp of either *B. brevicornis* or *B. hebetor* on varying numbers of *E. kuehniella* larvae was recorded. Therefore, one female wasp of each *Bracon* strain and 5, 10, 15, 20 or 25 *E. kuehniella* larvae, respectively, were kept in Petri dishes or plastic tubes with rectangular shape (5 cm diameter, 2.0 cm × 15.2 cm × 2.0 cm height × width × depth) in the climate chamber regimes as described above. Since female wasps have not been fed during the experiments, they were removed after 4 days. Furthermore, we expected 4 days to be the timeframe of the main oviposition period (*Kabore et al., 2019*). Larvae were tested for paralysis through a short squeeze of the head capsule with a featherweight forceps, showing no movement when paralyzed. Numbers of paralyzed and still active larvae were documented. Subsequently, paralyzed larvae were kept in climate chambers until all parasitoid larvae hatched but no longer than 5 weeks after the start of the experiment. Afterwards the samples were frozen and the number of hatched female and male wasps was documented. As a control, the same number of host larvae was incubated over the same period without female wasps, and moth development was documented. To determine the efficacy of pest control by the braconid wasps, paralyzation rates were calculated as paralyzation rate = number of paralyzed host larvae/ number of presented host larvae.

## Data analysis

Data were analysed with RStudio Version 1.1.383. First, the impact of strain affiliation was tested using linear model ANOVA. Since no differences between strains were observed (temperature*strain: $F_{3,617} = 0.688$ $p = 0.560$, strain: $F_{3,617} = 1.146$ $p = 0.330$), all strains were pooled for further analyses. Before statistical analyses, normal distribution of residuals was tested by Shapiro test and variance homogeneity by Levene's test

The response of total offspring of either *B. brevicornis* or *B. hebetor* hatched from one host larva from one female to changing temperatures was tested using general linear model (GLM) with poisson distribution and Chi-test. Tukey pairwise comparison was used subsequently. Similarly, sex ratio of offspring in response to different temperatures and *Bracon* species was tested using GLM with Poisson distribution (see Appendix 3 for raw data).

For testing the response of total offspring or sex ratio when different numbers of host larvae were presented, temperature, species and number of host larvae were included as variables in a GLM analysis with Poisson distribution and Chi-test. The number of hatched parasitoids (total, male, female) was square root transformed before statistical analysis. For pairwise comparisons, Tukey post hoc test was used (see Appendix 4 for raw data).

Differences in the efficacy of the braconid wasps between temperature and number of host larvae was tested using Kruskal–Wallis tests because of lacking normal distribution.

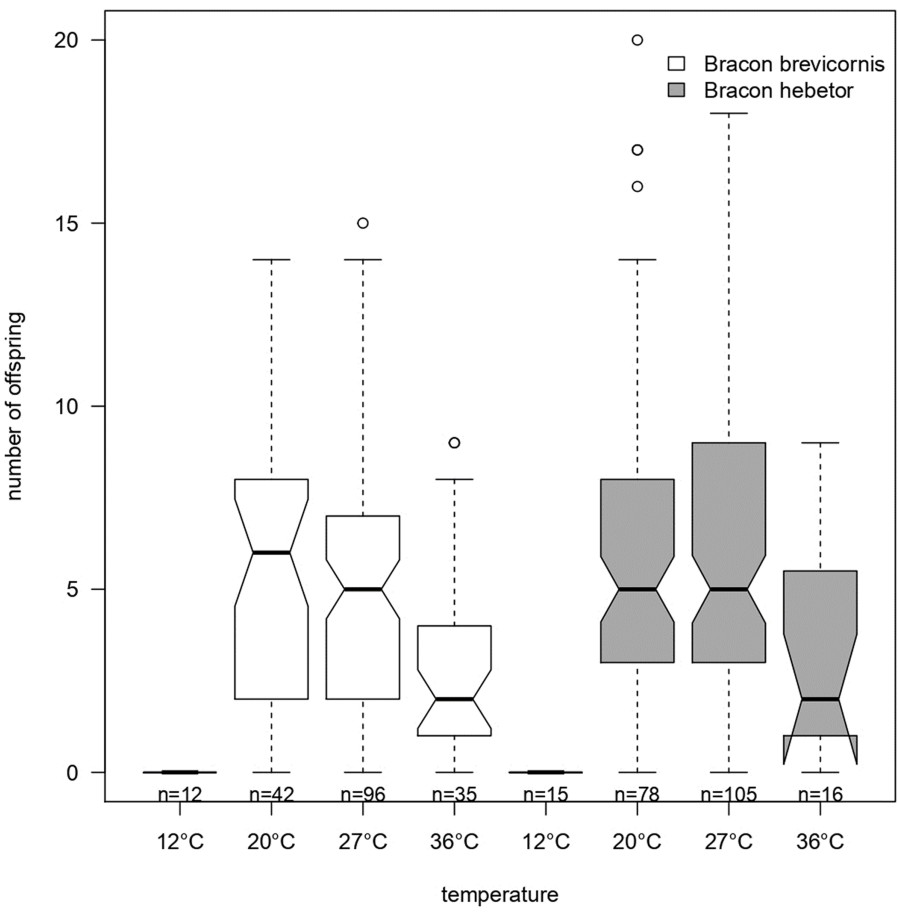

**Figure 2 Total number of offspring emerging from one host larva.** Total number of offspring emerging per host larva when *Bracon brevicornis* (white) or *Bracon hebetor* (dark grey) are each given one single host larva to parasitize at constant different temperatures (12 °C, 20 °C, 27 °C, 36 °C). Notches symbolize the 95% confidence. *n* = number of replicates.

## RESULTS

### How does the reproductive success of female wasps of *B. brevicornis* and *B. hebetor* on a single host larva change with temperature?

The reproductive success of *Bracon* females differed with temperature and species (temperature × species: $\chi^2_{(3,399)}$ = 21.90, $p$ = 0.046; species: $\chi^2_{(3,399)}$ = 21.90, $p$ < 0.001). In *B. brevicornis*, mean numbers of offspring decreased from ~8 at 20 °C and 27 °C to ~4 at 36 °C although the decrease was not significant (temperature: $\chi^2_{(1,185)}$ = 0.131, $p$ = 0.958). In *B. hebetor*, mean number of offspring significantly decreased from ~9 at 20 °C and 27 °C to ~5 at 36 °C (temperature: $\chi^2_{(1,214)}$ = 7.88, $p$ = 0.005). Both species failed to produce offspring at 12 °C and had the highest offspring at 20 °C and 27 °C (Fig. 2, Table 1).

In both species, the sex ratio was not temperature dependent (species × sex: $\chi^2_{(11,299)}$ = 272.57, $p$ = 0.105; temperature × sex: $\chi^2_{(11,299)}$ = 272.57, $p$ = 0.907). Although not significant, in *B. brevicornis*, the sex ratio was balanced at 20 °C (♂/♀ 1.02), became

**Table 1 Comparion of hatched total offspring from parasitization on one host larva at oviposition at different temperatures.** *P*-values of Tukey pairwise comparion of hatched total offspring from parasitization on one host larva at oviposition at different temperatures (12 °C, 20 °C, 27 °C, 36 °C) *in Bracon hebetor* and *Bracon brevicornis*. Significance value: *$p < 0.05$, **$p < 0.01$, ***$p < 0.001$.

| Comparsion | Bracon brevicornis | | Brevicornis hebetor | |
|---|---|---|---|---|
| 12 °C/20 °C | <0.001 | *** | <0.001 | *** |
| 12 °C/27 °C | <0.001 | *** | <0.001 | *** |
| 12 °C/36 °C | 0.033 | * | 0.044 | * |
| 20 °C/27 °C | ns | | ns | |
| 20 °C/36 °C | 0.002 | ** | 0.020 | * |
| 27 °C/36 °C | 0.001 | ** | 0.020 | * |

female biased at 27 °C (0.86), but was male biased at 36 °C (1.80). In *B. hebetor*, a male biased sex ratio was found at 20 °C and 36 °C (1.84 and 11, respectively), but it was about balanced at 27 °C (1.07).

## How does the reproductive success of female wasps of *B. brevicornis* and *B. hebetor* change with the number of available host larvae at different temperatures?

Presenting different numbers of host larvae at different temperatures to females of *B. brevicornis* and *B. hebetor* revealed a similar pattern of offspring for both species (host larvae × species × temperature: $\chi^2_{(7,627)} = 1073.78$, $p = 0.512$; Fig. 3, Table 2). In general, the number of offspring in *Bracon* species differed with the number of presented host larvae and temperature (host larvae respectively temperature: $\chi^2_{(7,627)} = 1073.78$, $p < 0.001$). At 12 °C, no offspring hatched at any number of presented host larvae, but most host larvae were paralyzed and did not develop to imagoes within 5 weeks. At 27 °C, the highest total offspring was observed in both species (Fig. 3, Appendix 5). At 36 °C, oviposition was observed in 20 of 140 samples, and only few parasitoids hatched; therefore, statistical analysis for hatched offspring within this treatment seemed inappropriate.

In *B. brevicornis*, the number of total offspring at 20 °C did not change with the number of presented larvae ($F_{1,98} = 0.05$, $p = 0.826$; Fig. 3A, Appendix 5). Mean values varied between 11.3 ± 6.6 at 15 host larvae and 19.4 ± 11.5 at 10 host larvae. At 27 °C, the number of total offspring varied significantly with the number of presented host larvae, changing from 27.9 ± 10.7 at 5 host larvae to 45.7 ± 19.3 on average at all higher numbers of host larvae ($F_{1,88} = 21.46$, $p < 0.001$; Fig. 3A, Appendix 5).

In *B. hebetor*, numbers of total offspring did not change significantly with numbers of host larvae at any temperature (host larvae: $\chi^2_{(3,321)} = 126.94$, $p = 0.177$; temperature: $\chi^2_{(3,321)} = 126.94$, $p = 0.138$) although mean numbers varied between 12.7 ± 5.9 at 10 host larvae and 20.8 ± 19.9 at 20 host larvae (Fig. 3B, Appendix 5).

Sex ratio of offspring varied marginally with numbers of host larvae between *Bracon* species (host larvae × species: $F_{1,247} = 3.78$, $p = 0.053$), but was independent of temperature ($F_{1,251} = 1.04$, $p = 0.308$; Fig. 4, Appendix 5). In general, fluctuations of sex ratios of *Bracon*

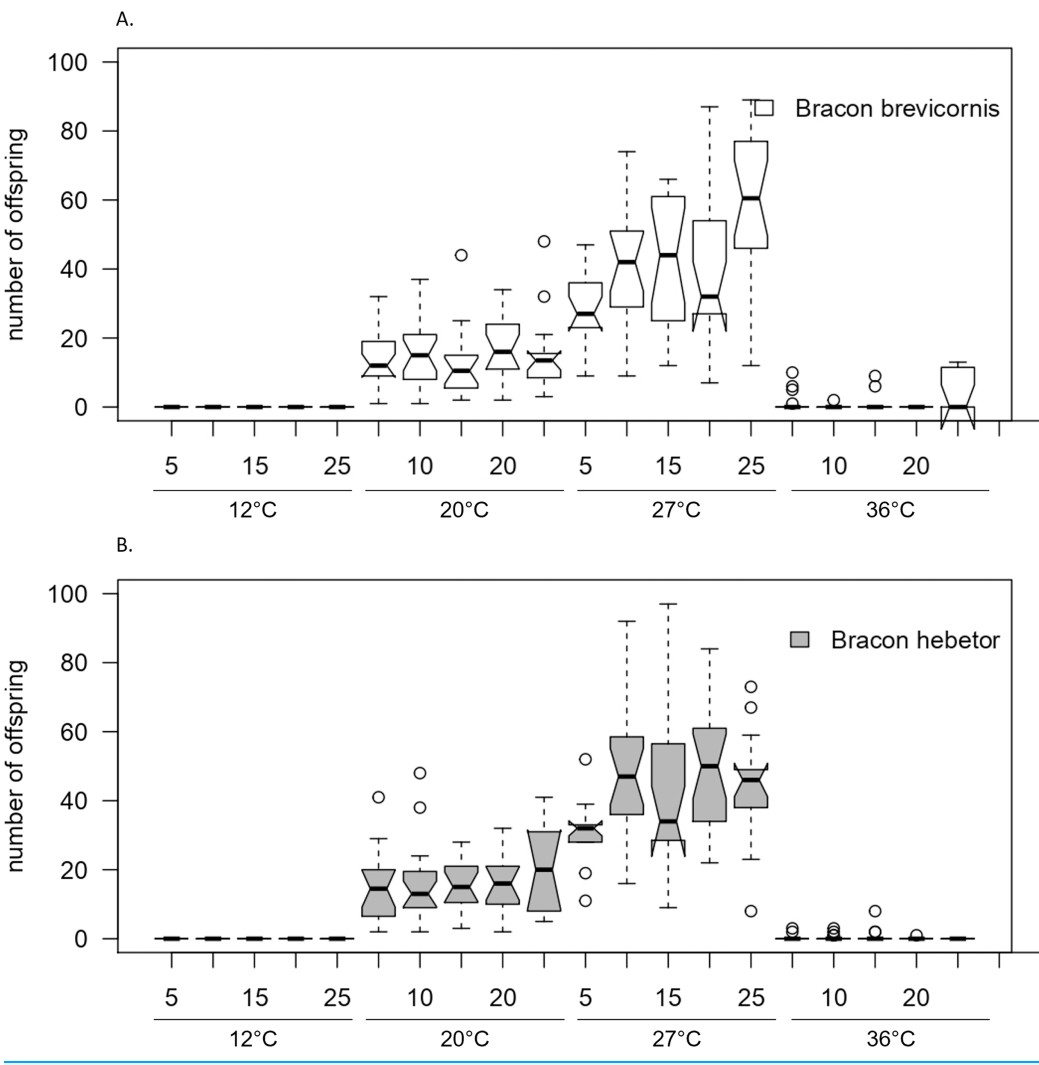

**Figure 3 Total number of offspring emerging from different numbers of host larvae at different temperatures.** Total number of offspring emerging per host larva when (A) *Bracon brevicornis* and (B) *Bracon hebetor* are each given 5, 10, 15, 20 or 25 host larvae to parasitize at different constant temperatures (12 °C, 20 °C, 27 °C, 36 °C) for 4 days. Notches symbolize the 95% confidence interval indicating significant differences among groups when no overlap occurs.

offspring were smaller at 27 °C compared to 20 °C and smaller in *B. hebetor* as compared to *B. brevicornis* (Fig. 4). While offspring was generally male biased, females dominated in *B. brevicornis* at 10 presented host larvae at 27 °C and in *B. hebetor* at 25 presented host larvae at 20 °C. A sex ratio of approximately 1:1 was observed at 15 host larvae at 20 °C in both *Bracon* species and at 15 host larvae at 27 °C in *B. brevicornis* (Fig. 4, Appendix 5).

Considering the offspring output per host larva, the highest number of offspring per larva was observed when only one host larva was present which was independent of the temperature (Table 3). If more than one host larva was present, a higher number of

**Table 2 Influence of different numbers of presented host larvae at different temperatures on the total offspring of *Bracon brevicornis* and *Bracon hebetor*.** Statistical results for general linear model analysis testing the influence of different numbers of presented host larvae at different temperatures on the total offspring of *Bracon brevicornis* and *Bracon hebetor*. Abbreviations: no., number; df, degree of freedom; resid, residuals; dev, deviance. Significances in bold: *$p < 0.05$, ***$p < 0.001$.

| | df | deviance | resid df | resid dev | F | p | |
|---|---|---|---|---|---|---|---|
| No. of host larvae | 4 | 347.8 | 625 | 16,569 | 13.947 | **<0.001** | *** |
| Species | 1 | 32.2 | 624 | 16,536 | 1.292 | 0.256 | |
| Temperature | 3 | 446.6 | 623 | 16,093 | 17.905 | **<0.001** | *** |
| No. host larvae*species | 4 | 58.6 | 622 | 16,031 | 2.349 | 0.126 | |
| No. host larvae*temperature | 12 | 62.2 | 621 | 15,969 | 2.494 | 0.115 | |
| Species*temperature | 3 | 115.6 | 620 | 15,853 | 4.636 | **0.032** | * |
| No. host larvae*species*temperature | 12 | 10.7 | 619 | 15,843 | 0.430 | 0.512 | |
| | | | 626 | 18,916 | | | |

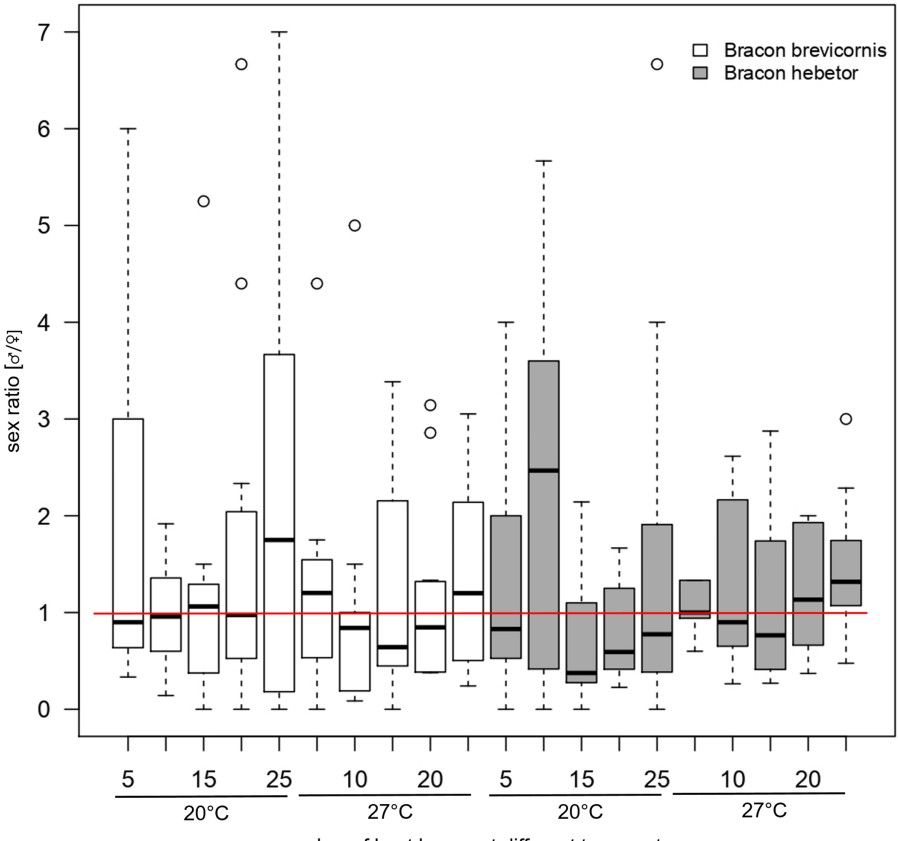

**Figure 4 Sex ratio (♂/♀) of offspring.** Sex ratio (♂/♀) of offspring emerging per host larva when *Bracon brevicornis* (white) and *Bracon hebetor* (dark grey) are each given 5, 10, 15, 20 or 25 host larvae to parasitize at different numbers of host larvae (5, 10, 15, 20 and 25) at different constant temperatures (12 °C, 20 °C, 27 °C, 36 °C) for 4 days. Red line marks an equal sex ratio.

**Table 3 Mean number of offspring per one host larva when emerged from different numbers of host larvae at different temperatures.** Mean number of offspring per one host larva when emerged from different numbers of host larvae (5, 10, 15, 20, 25) at different temperatures (20 °C, 27 °C) in *Bracon brevicornis* and *Bracon hebetor*. Standard deviations are given in brackets.

| No. of presented host larvae | *Bracon brevicornis* | | *Bracon hebetor* | |
|---|---|---|---|---|
| | 20 °C | 27 °C | 20 °C | 27 °C |
| 1 | 8.3 ± 4.1 | 8.2 ± 4.1 | 9.2 ± 4.8 | 9.1 ± 3.9 |
| 5 | 2.9 ± 1.5 | 5.6 ± 2.1 | 2.9 ± 2.0 | 6.2 ± 1.9 |
| 10 | 1.7 ± 1.1 | 4.1 ± 1.8 | 1.6 ± 1.1 | 4.9 ± 1.9 |
| 15 | 0.8 ± 0.7 | 2.8 ± 1.3 | 1.0 ± 0.5 | 2.8 ± 1.5 |
| 20 | 0.8 ± 0.4 | 2.0 ± 1.1 | 0.8 ± 0.4 | 2.5 ± 0.9 |
| 25 | 0.6 ± 0.4 | 2.4 ± 0.8 | 0.8 ± 0.5 | 1.8 ± 0.7 |

offspring per larva was observed at 27 °C in both *Bracon* species, being highest–although lower than on one host larva–when 5 or 10 host larvae were present (Table 3).

## Do different temperatures and varying numbers of host larvae affect the efficacy of braconid wasps?

Paralyzation rates differed significantly with temperature and species (temperature × species: $F_{2,641} = 7.14$, $p < 0.001$; Fig. 5, Table 4). In *B. brevicornis*, rates were continuously higher than 0.8 for all numbers of host larvae and temperatures. In *B. hebetor*, rates varied strongly between 0.5 at 12 °C with 10 or 15 host larvae and 1or almost 1 (0.99) at 36 °C for all numbers of host larvae (Fig. 5, Table 4).

## DISCUSSION

### How does the reproductive success of female wasps of *B. brevicornis* and *B. hebetor* on a single host larva change with temperature?

While presenting one host larva to one female wasp in our study, *B. hebetor* produced slightly higher numbers of offspring compared to *B. brevicornis* and oviposition was reduced by increasing temperature. *Yu et al. (2003)* reported an average of 12.8 eggs oviposited by one *B. hebetor* female on one host larva (*Plodia interpunctella*, 28 ± 0.5 °C, 70–75% relative humidity, 16:8 L:D photoperiod). Since the average numbers of eggs per one host larvae in our study were lower (~9 at 20 °C and 27 °C), the choice of the host species seems to have an impact on the reproductive outcome. Previous studies reported that the best host species for rearing *Bracon* wasps in the laboratory is the greater wax moth *Galleria mellonella* (*Farag et al., 2015*; *Khalil et al., 2016*) which can produce about 260 total offspring within about 38 days (*Nikam & Pawar, 1993*).

However, highest number of females, which are needed to control host pest species, were produced at 27 °C, with a sex ratio of nearly 1:1 in *B. hebetor* and even 0.86 in *B. brevicornis* while nearly all other temperatures showed male-biased sex ratios. Since only female wasps contribute to the effective success of biological pest control, the sex ratio of a laboratory strain of *Bracon* wasps is of critical importance. *Mohamad, Mansour & Ramadan (2015)* reported a correlation between sex ratio and temperature whereby high

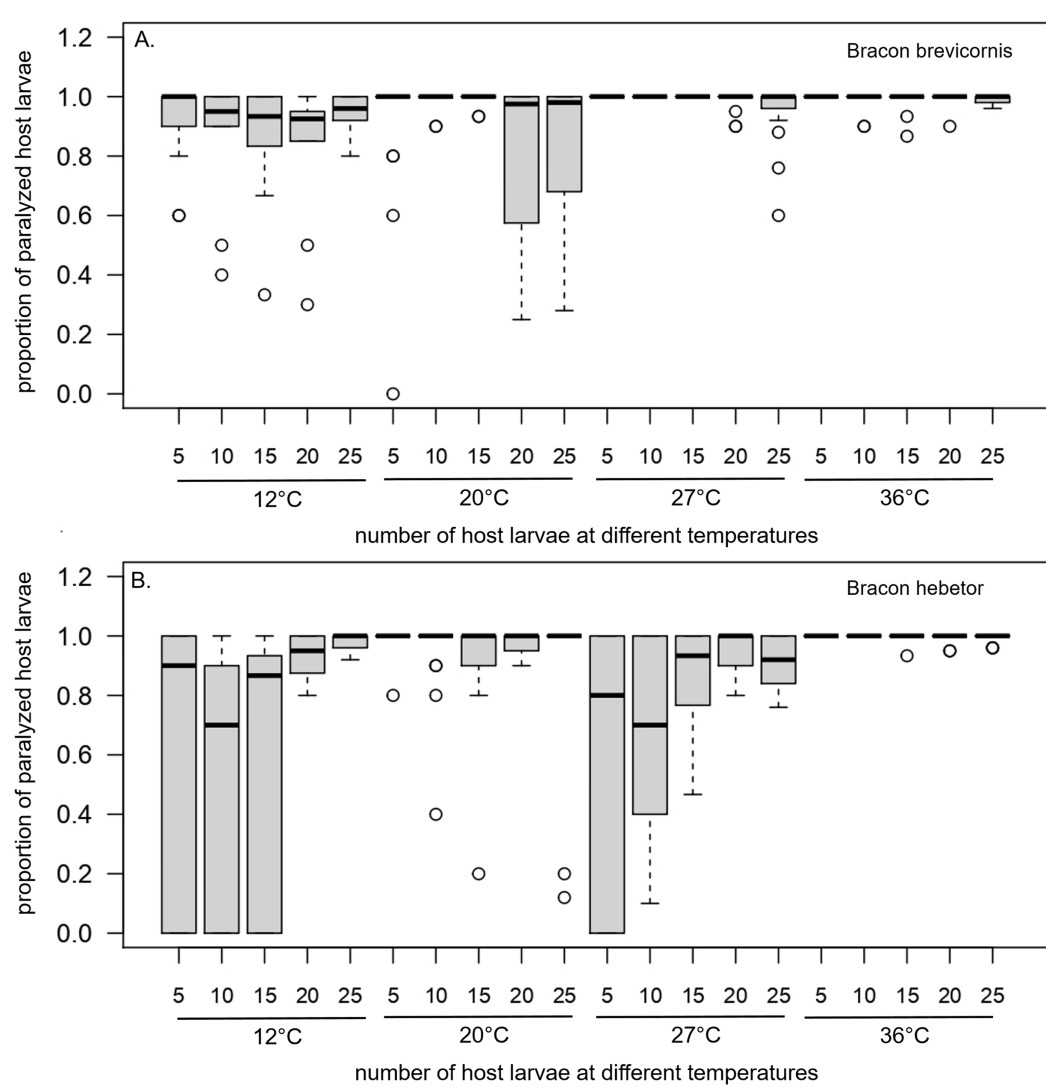

**Figure 5 Parasitization of host larvae.** Parasitization of host larvae when (A) *Bracon brevicornis* and (B) *Bracon hebetor* are each given 5, 10, 15, 20 or 25 host larvae to parasitize at different constant temperatures (12 °C, 20 °C, 27 °C, 36 °C) for 4 days.

**Table 4 Parasitization rates at different temperatures with varying numbers of host larvae.** Parasitization rates at different temperatures (12 °C, 20 °C, 27 °C, 36 °C) with varying numbers of host larvae (5, 10, 15, 20, 25). Standard deviations are given in brackets.

| | *Bracon brevicornis* | | | | *Bracon hebetor* | | | |
|---|---|---|---|---|---|---|---|---|
| | **12 °C** | **20 °C** | **27 °C** | **36 °C** | **12 °C** | **20 °C** | **27 °C** | **36 °C** |
| **5** | 0.93 ± 0.14 | 0.91 ± 0.24 | 1 ± 0 | 1 ± 0 | 0.63 ± 0.46 | 0.99 ± 0.05 | 0.63 ± 0.44 | 1 ± 0 |
| **10** | 0.89 ± 0.18 | 0.99 ± 0.03 | 1 ± 0 | 0.98 ± 0.04 | 0.52 ± 0.44 | 0.95 ± 0.14 | 0.66 ± 0.32 | 1 ± 0 |
| **15** | 0.86 ± 0.18 | 0.99 ± 0.02 | 1 ± 0 | 0.98 ± 0.04 | 0.57 ± 0.47 | 0.90 ± 0.20 | 0.87 ± 0.17 | 1 ± 0.02 |
| **20** | 0.84 ± 0.22 | 0.82 ± 0.27 | 0.99 ± 0.03 | 0.99 ± 0.04 | 0.93 ± 0.09 | 0.98 ± 0.03 | 0.95 ± 0.07 | 0.99 ± 0.02 |
| **25** | 0.93 ± 0.08 | 0.84 ± 0.23 | 0.95 ± 0.10 | 0.99 ± 0.02 | 0.98 ± 0.03 | 0.83 ± 0.35 | 0.91 ± 0.09 | 0.99 ± 0.02 |

temperatures (35 °C) resulted in higher male-biased offspring. In summary, for rearing *B. brevicornis* and *B. hebetor* in the laboratory on one host larva of *E. kuehniella* (but carefully consider host choice), a temperature between 20–27 °C seems most appropriate.

### How does the reproductive success of female wasps of *B. brevicornis* and *B. hebetor* change with the number of available host larvae at different temperatures?

The production of offspring and the pattern of sex ratios in response to different numbers of host larvae and temperature was very similar between *B. brevicornis* and *B. hebetor*. Presenting more than one host larva to one female wasp resulted in higher numbers of total offspring while the average number per larva was lower as when a single larva was present. Since host quality affects different performance parameters of parasitoids (*Bernal, Luck & Morse, 1998*; *Mody et al., 2017*), reducing the number of eggs per host larva could result in increased fitness per parasitoid larva due to reduced competition for food. Furthermore, the processing time one female wasp needs to detect and paralyze a host and to oviposit eggs increases with the number of hosts presented, resulting in less eggs in average (*Yu et al., 2003*).

As seen for only one host larva, highest numbers of offspring were observed at 27 °C in both species when more larvae were present. Higher temperatures (36 °C) seem to negatively affect the development, probably by heat stress. Direct effects of heat stress were reported by *Klockmann, Kleinschmidt & Fischer (2017)*, showing that heat stress reduced survival rates and fitness of a Lepidoptera larva (*Bicyclus anynana*). Furthermore, there are indirect effects of heat stress for parasitoids (*Singh, Singh & Tripathi, 2014*). *Bracon* wasps belong to the holometabolous insects that undergo a metamorphosis from egg over larva and pupa to adult. For the metamorphosis and the pupation of the larvae the so-called 'critical weight' must be achieved. When the critical weight is attained, the larvae begin to pupate and start the metamorphosis (*Harrison, Woods & Roberts, 2012*). At 36 °C, the loss of water of host larvae may cause a reduction of the food quality for the parasitoid larvae. If parasitoid larvae starve, the critical weight is not reached; therefore, the larvae cannot develop into an adult but die. On the other hand, no oviposition and nearly no activity of the female wasps was observed at 12 °C, indicating that they entered chill-coma, a reversible state that is caused by a reduction of physiological processes (*MacMillan & Sinclair, 2011*; *Singh, Singh & Tripathi, 2014*).

Besides temperature, braconid wasps showed adjustments in their oviposition behavior according to differences in host density in previous studies (e.g., *Godfray, 1994*; *Yu et al., 2003*; *Milonas, 2005*; *Mohamad, Mansour & Ramadan, 2015*). In *B. brevicornis*, numbers of offspring increased with increasing numbers of presented host larvae at 27 °C. In *B. hebetor* on *Corcyra cephalonica*, searching efficiency of female parasitoids was best when five host larvae were presented (*Singh, Singh & Tripathi, 2016*).

In this study, this pattern could be confirmed when *Bracon* species were reared on one host larva only, but when more host larvae were present the sex ratio of offspring was independent from temperature. On the other hand, sex ratio of offspring slightly changed with varying numbers of host larvae. While *Yu et al. (2003)* found a sex ratio of

approximately 0.5 irrespective of the number of presented host larvae, other studies reported labile sex ratios of *B. hebetor* in response to density, but with different results: if parasitoid densities were high more females were produced (*Galloway & Grant, 1988*), but on *Galleria mellonella* the proportion of females decreased with increasing parasitoid density (*Alam et al., 2016*).

## Do different temperatures and varying numbers of host larvae affect the efficacy of braconid wasps?

Practitioners like farmers care about the efficacy of pest control and not about the number of offspring of the control agent. Nevertheless, they benefit from an economical breeding process, as this allows lower prices per agent application. To replace or at least supplement pesticides, biological control agents should be nearly as effective as pesticides but provide additional advantages.

The current study showed high effectiveness of *Bracon* species at all temperatures (12–36 °C) and host numbers (five-25) with high paralyzation rates under laboratory conditions. However, concerning the efficacy both *Bracon* species showed the greatest differences. While *B. brevicornis* was effective at all temperatures, *B. hebetor* was less effective at low temperatures. However, although no offspring was observed at 12 °C, female wasps must have injected venom. Since female wasps were kept at room temperature before they were confronted with the host larvae and exposed to 12 °C, they could have paralyzed the host larvae before turning to the chill coma. The paralysis and efficacy at low temperatures represent particular advantages for practitioners. When pest-infested grain- or other stored-products are chilled and not frozen during storage, pests turn over to chill coma, but can develop when conditions become favorable again (*Evans, 1987*; *Locatelli, Papale & Daolio, 1990*; *Andreadis, Eliopoulos & Savopoulou-Soultani, 2012*). Preventive usage of *B. brevicornis* or *B. hebetor* could protect grain- or other stored-products before and during chilling against pests like *Ephestia kuehniella* or *Plodia interpunctella*.

Furthermore, parasitization over a broad range of temperatures and host densities makes both *Bracon* species more effective for their use in open fields, where environmental conditions like temperature can vary. In Germany, there are guidelines ('good agricultural practices') regulating the use of pesticides in connection with environmental conditions. Pesticide use should hence be avoided when temperatures permanently exceed 25 °C, which becomes more common in summer, or air humidity is below 30%, otherwise penalties will have to be paid. *B. brevicornis* or *B. hebetor* could be used effectively in either case.

To summarize the useful facts of this study for breeders: using ten host larvae and one female wasp at 27 °C is recommended to increase the number of offspring in total and to keep the number of hatched parasitoids per host at a high level. Practitioners can use either *B. brevicornis* or *B. hebetor* at low and high temperatures as well as with varying host densities to achieve a high level of effectiveness of pest control.

## ACKNOWLEDGEMENTS

The authors thank AMW Nützlinge from Pfungstadt for providing the first generation of the parasitoids and providing hosts regularly. We further thank Nadja Simons for supporting the statistical analyses.

### Funding

We received support from the Open Access Publishing Fund of Technical University of Darmstadt.

### Grant Disclosures

The following grant information was disclosed by the authors:
Technical University of Darmstadt.

### Competing Interests

Tore-Aliocha Kursch-Metz is employed by AMW Nützlinge GmbH.

### Author Contributions

- Jessica Lettmann conceived and designed the experiments, performed the experiments, analyzed the data, prepared figures and/or tables, authored or reviewed drafts of the paper, and approved the final draft.
- Karsten Mody conceived and designed the experiments, analyzed the data, authored or reviewed drafts of the paper, and approved the final draft.
- Tore-Aliocha Kursch-Metz conceived and designed the experiments, performed the experiments, authored or reviewed drafts of the paper, and approved the final draft.
- Nico Blüthgen conceived and designed the experiments, analyzed the data, authored or reviewed drafts of the paper, and approved the final draft.
- Katja Wehner conceived and designed the experiments, analyzed the data, prepared figures and/or tables, authored or reviewed drafts of the paper, and approved the final draft.

### Data Availability

  The raw measurements are available in the Supplemental Files.

### Supplemental Information

Supplemental information for this article can be found online at http://dx.doi.org/10.7717/peerj.11540#supplemental-information.

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
