# Peer review of "Bracon wasps for ecological pest control–a laboratory experiment"

_PeerJ, doi:10.7717/peerj.11540_

## Round 0.1 · original submission · Major Revisions

Dear Dr. Lettmann and colleagues:

Thanks for submitting your manuscript to PeerJ. I have now received two independent reviews of your work, and as you will see, the reviewers raised some concerns about the research. Despite this, these reviewers are optimistic about your work and the potential impact it will have on research studying utilization of Bracon wasps for pest control. Thus, I encourage you to revise your manuscript, accordingly, taking into account all of the concerns raised by both reviewers.

The main concerns are a lack of sufficient background, missing references, incomplete descriptions of methods, and numerous edits to the text. Your work should be repeatable, so strive to provide everything necessary such that all assays can be repeated independently by others.

Please note that both reviewers have included marked-up versions of your manuscript.

There are many minor suggestions to improve the manuscript.

Therefore, I am recommending that you revise your manuscript, accordingly, taking into account all of the issues raised by the reviewers.

Good luck with your revision,

-joe

Reviewer 1 ·

Basic reporting

The paper is written in clear and professional English. There are some sentences that I believe should be reworded, which I noted in the annotated pdf. Otherwise, the paper is well written.
Most of the relevant literature is reported however, I believe two papers on the temperature effects on B. brevicornis, which I think should be cited since they are very similar to the study conducted.
Rao, B. R. S., and S. S. Kumar. 1960. Effect of Temperature and Host Density on the Rate of Increase of Bracon brevicornis Wesmael (Hymenoptera: Braconidae). Beiträge zur Entomol. 10: 872–885.
Thanavendan, G., and S. Jeyarani. 2010. Effect of different temperature regimes on the biology of Bracon brevicornis Wesmael (Braconidae: Hymenoptera) on different host larvae. J. Biopestic. 3: 441–444.
The article is well structured, and the hypotheses are clearly stated and supported by data.
I only have two comments on the raw data provided.
Supplemental data:
Appendix 1 Strains: Specify where the B. brevicornis was caught with a hand net if possible
Appendix 2: Unclear what ges stands for.
I have a few comments on the figures and tables, which I noted in the annotated pdf. My major comment is that I do not believe both tables and graphs should be included in the main paper as they represent similar data.
The paper is self-contained, and the results are relevant to the hypotheses.

Experimental design

This is original primary research on the B. brevicornis and B hebetor parasitoids reared on Ephestia kuehniella. However, to improve the introduction and conclusion clearly emphasize why E. kuehniella is important, and why temperature and host density are important for parasitoid rearing.
The methods are described in sufficient detail and information to replicate. Methods are also typical of those found in other parasitoid rearing papers. The investigation had sufficient replication and was performed at a high technical and ethical standard.

Validity of the findings

All underlying data have been provided, and statistics are appropriate for this study. Conclusions are well stated and linked to original research questions.
Results support the majority of the discussions and conclusions, however, there is some speculation on how these results can be used for the rearing of the Bracon species on other hosts. This final speculation on lines 331- 336 is not identified as being such.

Additional comments

I made suggestions for minor improvements in the annotated pdf.

Annotated reviews are not available for download in order to protect the identity of reviewers who chose to remain anonymous.

Reviewer 2 ·

Basic reporting

The paper perfectly falls within the scope of the journal and deserves publications, provided some improvement. I will recommend getting the English of the manuscript thoroughly edited.
I have made some suggestion in the PDF but they are not exhaustive.
The introduction should cover some basic on the two parasitoid wasps biology such as temperature requirement and life table. When referring to host species of Bracon hebetor you should include Corcyra cephalonica which is widely used in Africa and India. It is a bit surprising you missed C. cephalonica while Ostrinia nubilalis which is not among most common host species is cited.
As a general rule, make sure that when referring to different organisms for the first time in the manuscript to use the recommended nomenclature (e.g. Bracon hebetor Say). It will also be good to mention that Bracon (= Habrobracon) hebetor.
-A short description of the two Bracon species is needed as B. hebetor and B. brevicornis are sometime being confused.
-Regarding Tables and Figures, I’ve had the impression that the data use to generate table 1 where the same used in Figure 1. Why do you need to present total number of emergence in figure 1 and them break it per sex in table 1 while presenting the sex ratio in the same table. This is redundant. The only new data in Table 1 as compare to Figure 1 is the sex ratio. Please discard table 1 and present the sex ratio in the text while commenting the result on offspring emerging from single host-larva parasitism. The same comment may apply to Table 4. Please check other tables to avoid duplication of data presented in figures.
-Figures and tables captions must be revised as one should be able to understand what the Figure/table upfront. I have made some suggestion (see attachment)

Experimental design

In the experimental section it is stated that after five days, the generation was frozen and newly hatched wasps were used. This is unclear, please explain clearly what you meant.
The experiment on reproductive success of female wasps on a single host larva at different temperatures is not well described. It is unclear for how long Bracon was given the host larva to parasitize. It is also unclear what the replicate stands for, is it 1 female of a given species at a different temperature. You should mention clearly the number of replicates you used while starting the experiment. I can understand that at the end some replicates are not considered for one of the reasons that you mentioned (survival rate), but I do not understand why you discarded some replicates because of delayed in the parasitoid developmental cycle.
While testing Bracon with different host larvae, It is unclear how many replicates were used for each species and strain at a given host-ratio (1:5; 1:10; …1:25) and given temperature (12 to 36°C). It would have also been good to know the rationale behind the number of host larvae (5 to 25) that you choose to test. Likewise, what is the rationale for confining host larvae with Bracon parasitoid for 4 days, why not 48hrs. These questions (host exposure duration and host-parasitoid ratio) could have been avoided if enough background information was given in the M&M section while describing the two species
-In Figure 3 & 4 it appears that you didn’t test the same number of host-larva at different temperatures (5,15, 25 larvae at 12C and 27C and 10, 20 larvae at 20C and 36C). This should be explained in the Material and Method.
-I do not understand why parasitism of Bracon in figure 5 is not expressed as percentage. This should be corrected.

Validity of the findings

The findings are good but not always well presented.
-There is a need to discuss the findings in relation to previous one reporting 200-400 offspring produced per Bracon hebetor female in other settings with different host species (see Kabore et al., 2019; Nikam and Pawar 1993; Yu et al. 1999; Chen et al. 2011). Likewise, when comparing the performance of Bracon with other studies it should be good to consider the diet on which the host-larva was reared as this could influence parasitism and progeny production (see Amadou et al. 2019). By the way this is missing in the M&M section.
In the conclusion it is suggested to test the usefulness of Bracon wasps in controlling insect pests in field situation. This should not even be questioned given that Bracon hebetor is already used in augmentative biological control program in Asia and Africa for several insect pests (search literature on Bracon hebetor against millet head miner in Africa and Bracon hebetor against Carob moth in Iran).

Here is a list a reference that you find useful:
Yu S-H, Ryoo MI, Na JH (1999) Life history of Bracon hebetor (Hymenoptera: Braconidae) on Plodia interpunctella (Lepidoptera: Pyralidae) on a dried vegetable commodity. J Asia Pac Entomol 2: 149–152

Nikam PK, Pawar CV (1993) Life tables and intrinsic rate of natural increase of Bracon hehetor say (Hym., Braconidae) population on Corcyra cephalonica Staint. (Lep., Pyralidae), a key parasitoid of Helicoverpa armigera Hbn. (Lep., Noctuidae). J Appl Entomol 115: 210–213

Chen H, Opit GP, Sheng P, Zhang H (2011) Maternal and progeny quality of Habrobracon hebetor say (Hymenoptera: Braconidae) after cold storage. Biol Control 58:255–261

Amadou L, Baoua IB, Ba NM, Muniappan R (2019) Develop- ment of an optimum diet for mass rearing of the rice moth, Corcyra cephalonica (Lepidoptera: Pyralidae), and pro- duction of the parasitoid, Habrobracon hebetor (Hy- menoptera: Braconidae), for the control of pearl millet head miner. J Insect Sci 19:1–5

Kabore, A., M. N. Ba, C. Dabire-Binso, A. Sanon. 2019. Towards development of a parasitoid cottage industry of Habrobracon hebetor (Say): optimum rearing and releases conditions for successful biological control of the millet head miner Heliocheilus albipunctella (De Joannis) in the Sahel. Int. J. Trop. Insect Sci. doi: 10.1007/s42690-019-00005-w

Annotated reviews are not available for download in order to protect the identity of reviewers who chose to remain anonymous.

---

## Round 0.2 · Minor Revisions

Dear Dr. Lettmann and colleagues:

Thanks for revising your manuscript. The reviewers are very satisfied with your revision (as am I). Great! However, there are a few issues to entertain. Please address these ASAP so we may move towards acceptance of your work. Please note that reviewer 2 has included a marked-up version of your manuscript.

Best,

-joe

Reviewer 1 ·

Basic reporting

no comment

Experimental design

no comment

Validity of the findings

no comment

Additional comments

The authors have addressed all of my concerns.

Reviewer 2 ·

Basic reporting

The Manuscript has been significantly improved and reads better. My recommendations have all been taken into consideration. I have however made few suggestions in the PDF for

-As a general rule always give Order and Family of organism when citing them for the first time in the text (e.g. Ostrinia nubilalis Hübner (Lepidoptera: Crambidae); Orius insidiosus Say (Hemiptera: Anthocoridae). Please check for other organisms in the attachment.

-Regarding Tables and Figures, I’ve had the impression that the data use to generate table 4 where the same used in Figure 5.

-Check attachment for further suggestions

Experimental design

-It is important to indicate initial number of replications thought some were discarded due to failure.
- I do not agree with the method use for computing parasitism. Unless you have supporting reference, a parasitized larva is a host larva on which the parasitoid lay eggs. Bracon can sting and paralyze the host larvae without laying egg on it (see Kabore et al., 2019). The host-larvae can eventually be killed as a result of host feeding but not due to parasitism. In your case host feeding may have been important given that parasitoids were not given any food (honey, sweet solution…)

Validity of the findings

My previous comments have all been addressed. However I am suggesting within the discussion to consolidate all the section related to sex ratio in one paragraph.

Annotated reviews are not available for download in order to protect the identity of reviewers who chose to remain anonymous.

---

## Round 0.3 · accepted · Accept

Dear Dr. Lettmann and colleagues:

Thanks for revising your manuscript based on the concerns raised by the reviewers. I now believe that your manuscript is suitable for publication. Congratulations! I look forward to seeing this work in print, and I anticipate it being an important resource for groups studying the utilization of Bracon wasps for pest control. Thanks again for choosing PeerJ to publish such important work.

Best,

-joe